# The Dual Pathogen *Fusarium*: Diseases, Incidence, Azole Resistance, and Biofilms

**DOI:** 10.3390/jof11040294

**Published:** 2025-04-09

**Authors:** Dongmei Li, Kincer Amburgey-Crovetti, Emilie Applebach, Tomoko Y. Steen, Richard Calderone

**Affiliations:** Department of Microbiology and Immunology, School of Medicine, Georgetown University, 3900 Reservoir Rd., Washington, DC 20057, USA; ka960@georgetown.edu (K.A.-C.); ema122@georgetown.edu (E.A.); tys8@georgetown.edu (T.Y.S.); calderor@georgetown.edu (R.C.)

**Keywords:** *Fusarium* species complexes, fusariosis, biofilm biology, antifungal resistance

## Abstract

The increasing resistance of *Fusarium* species to nearly all first-line antifungal agents in clinical settings has led to its designation as a ‘high-priority’ human pathogen. As a dual pathogen, *Fusarium* spp. threaten both human health and crop production, impacting food security. Our recent drug profiling of clinical *Fusarium* isolates reveals resistance to several front-line antifungals, with notable cross-azole resistance observed in both clinical and plant-associated strains. While the overuse of agricultural azoles has been implicated in the selection of azole-resistant fungi such as *Aspergillus*, a similar mechanism has been assumed for *Fusarium* in clinical settings. However, direct genetic evidence supporting this hypothesis remains limited. In this review, part of our Special Interest (SI) series, we discuss the spectrum of human diseases caused by *Fusarium*. While incidence data are better established for human keratitis and onychomycosis, invasive fusariosis remains globally underreported. We propose reasons for this distinct clinical spectrum bias and explore the potential genetic basis of azole resistance.

## 1. Introduction

*Fusarium* species belong to a group of dual-threat, cross-kingdom, filamentous fungal pathogens that cause both human infections and disruptions in food security and supply [1,2], designated as a ‘high-priority’ human pathogen [3,4]. Over 70% of all clinical *Fusarium* isolates are thought to arise through contact with airborne or waterborne spores in the environment [5]. While several filamentous fungi share these characteristics, *Fusarium* and *Aspergillus* species are among the most frequently encountered [6]. A significant number of hospitalized patients with resistant *Fusarium* and *Aspergillus* fungemia have no history of azole use, suggesting an environmental or airborne hospital source of resistance [7,8]. The emergence of environmentally driven azole resistance presents a major challenge in clinical treatment. This phenomenon is well documented in *Aspergillus fumigatus*, where resistance is linked to characteristic point and promoter mutations in the azole target gene, lanosterol 14α-demethylase (*CYP51*) [7]. However, the similar evidence for *Fusarium* in human infections remains unconfirmed.

The prevalence of the cross-resistance between medical and agricultural azoles in clinical settings may be due to their shared target protein. Currently, we still lack molecular markers to distinguish whether airborne *Fusarium* spores that infect hospitalized patients originate from hospital environments or external sources. Both the molecular mechanisms of resistance and the extent of *Fusarium* resistance remain significantly understudied, particularly in the United States. To date, only one study has reported the azole-resistant mechanism of *Fusarium*, involving 15 isolates from clinical and environmental sources in Malaysia [9].

Taxonomically, *Fusarium* comprises over 300 phylogenetically distinct species, grouped into more than 20 species complexes, most of which are found in the environment [8,10]. The majority of medically significant *Fusarium* species belong to seven key species complexes: *Fusarium solani* species complex (FSSC), *Fusarium oxysporum* species complex (FOSC), *Fusarium fujikuroi* species complex (FSSC), *Fusarium incarnatum-equiseti* species complex (FIESC), *Fusarium chlamidosporum* species complex (FCSC), *Fusarium dimerum* species complex (FDSC), and *Fusarium sporotrichoides* species complex (FSAMSC) [8,10].

## 2. Clinical Types and Incidence of Human Fusariosis

Human fusariosis is a reemergent worldwide disease exhibiting marked increases since the 1970s–1980s, according to Lockhart and Guarner [11]. Human infections caused by *Fusarium* fall into three major categories: superficial infections such as (1) keratitis, (2) onychomycosis (nail infections), and (3) invasive infections, including fungemia and disseminated fusariosis occurring in immunocompromised patients [8,12]. Disseminated infection in patients with fusariosis typically affects the skin in over 50% of cases [13], with deep organ involvement, such as pneumonia or fungemia, associated with poorer outcomes [12,14]. In contrast, localized infections like corneal keratitis [15,16] and onychomycosis (nail infections) [17] are more commonly observed in immunocompetent individuals. Despite advances in treatment, the effectiveness of interventions for both invasive and localized fusariosis remains suboptimal. Major risk factors for fusariosis include invasive surgery, organ transplants, chronic steroid use, and trauma to the skin, nails, and eyes [8,9,11].

### 2.1. Keratitis

*Fusarium* keratitis is the most common clinical manifestation of fusariosis [15,18]. A 10-year study of *Fusarium* infections at a French university hospital found that *Fusarium* cultures were isolated from 31% of ophthalmic samples, 8.48% of nail samples, and 0.47% of blood cultures [19]. Estimates from 43 published studies on country-level disease burdens, compiled by GAFFI, 2022 (Global Action For Fungal Infections, https://gaffi.org/media/country-fungal-disease-burdens, accessed on 4 February 2025), indicate that approximately 1,000,000 cases of fungal keratitis occur annually, which was adjusted to up to 1,400,000 patients each year [20]. In underdeveloped countries, these infections can lead to a high proportion of enucleation (11%) and visual loss (~50%) [15,21]. In tropical areas, *Fusarium* is the leading fungal pathogen responsible for fungal keratitis, in contrast to yeast fungal pathogens in subtropical areas [22].

Notably, *Fusarium* accounts for approximately 45–53% of all fungal keratitis cases, surpassing *Aspergillus* 3.3-fold [23]. Efforts have been made to classify fungal keratitis as a Neglected Tropical Disease (NTD) due to its severity and high prevalence in tropical and subtropical regions, particularly following corneal trauma [20]. Reports indicate that *Fusarium* accounts for 37.7% to 81.5% of all culture-positive corneal infections in these regions, often leading to severe visual impairment. However, *Fusarium* keratitis is not restricted to tropical climates—an outbreak in the United States linked 164 confirmed keratitis cases across 33 states and one U.S. territory to contaminated contact lens solution [24]. Additionally, cases of *Fusarium* keratitis have been reported following corneal transplant [18]. The relatively low incidence of *Fusarium* fungemia compared to its high prevalence in superficial infections highlights the need for better treatment strategies to reduce chronicity and potential latency [20]. Currently, voriconazole (VCZ) is still the most commonly used drug in *Fusarium* keratitis. Topical use of natamycin (NAT), amphotericin B (AMB), or a combination of both, is a better treatment than terbinafine, which showed a certain level of susceptibility in clinical *Fusarium* isolates [25,26].

### 2.2. Onychomycosis

What stands out when describing this disease is its chronicity (recurrence, in 2–3 years) and treatment failures [27]. Risk factors include aging, sex, genetic predisposition, diabetes, occlusive footwear, and nail trauma [28]. A recent study on fungal isolation from toenail samples in the USA identified *Fusarium* as the leading non-dermatophyte mold (NDM) in male patients [29]. Several factors may contribute to poor cure rates and the high prevalence of *Fusarium* in nail infections. First, *Fusarium* spp. are intrinsically resistant to azoles and other antifungal therapies. Onychomycosis caused by de novo dermatophyte infections typically requires antifungal treatment for three months (fingernails) or four months (toenails). Such prolonged treatment may promote the selection of resistant fungal pathogens, either as a sole infection or in combination with dermatophytes [30,31,32]. Although the exact prevalence of *Fusarium* co-infection alongside dermatophytes remains unclear, estimates range from approximately 5–30%. A protracted course of nail infection may warrant reevaluating the dominant fungal species involved.

Second, biofilm formation within nail tissue beyond the nail structure itself may hinder antifungal penetration, further complicating treatment. This highlights the need for additional research on the role of biofilms in onychomycosis and their impact on antifungal efficacy [33,34]. The role of biofilms in *Fusarium* infections will be discussed further in the section on biofilms. Finally, the emergence of NDM in onychomycosis raises concerns about its potential as a latent source of infection [31], since animal models have demonstrated that *F. oxysporum* can persist in immunocompetent hosts and is able to provoke the systemic infection upon immunosuppressive treatment [35].

### 2.3. Invasive Fusariosis (IF)

Invasive fusariosis often occurs when airborne *Fusarium* spores reach the alveoli, germinate, and form hyphae, leading to tissue invasion. The risk and severity of IF are quite sensitive to the patient’s immune status. For example, immunocompromised individuals with hematological malignancies, neutropenia, or solid organ transplants show far more severe consequences of exposure.

Historically, IF was considered a rare complication in hematological diseases. In prospective studies conducted between 1981 and 1996, the global incidence was reported at only 5.1 to 6.3 cases per year, with approximately half of the cases occurring in the United States [36,37]. However, by the 2010s, its incidence had increased among Brazilian patients with hematologic malignancies and allogeneic hematopoietic cell transplant (HCT) recipients, reaching 3.8–5.2% [38,39]. A 2019 report further highlighted this growing burden, documenting an incidence of 148 cases per 1000 patients with acute lymphoid leukemia and 13.1 cases per 1000 patients with myeloid leukemia [40]. The overall mortality rate of IF is 66%, rising to 96–100% in patients with persistent neutropenia [11].

Global incidence data on IF remain limited due to the lack of comprehensive country-wide reports for most countries. Instead, hospital-based studies must be used to expose the trend. For instance, Pérez-Nadales et al. [41] reported that IF-related hospital admissions per 100,000 population increased from 0.9 (2000–2009) to 2.2 (2009–2015) compared to the period before 2000. In the same study, the 90-day mortality rate was 91% among neutropenic patients, compared to 28.1% in non-neutropenic patients. Additionally, a separate study of 55 ICU patients in French hospitals found that 56–76% of those with IF developed pneumonia [42].

Laboratory diagnosis of IF includes blood cultures and histopathological examination of fungal elements in infected tissues, such as skin and lung lesional biopsies. In clinical settings, diagnostic assays include MALDI-TOF (Mass Spectrometry Matrix-Assisted Laser Desorption-Ionization) mass spectrometry, β-D-glucan assays, and the galactomannan (GM) test. MALDI-TOF identifies microbial species based on unique protein profiles [43], while β-D-glucan and GM assays detect fungal cell wall components. The GM assay, originally developed for *Aspergillus* [44], showed low sensitivity in IF, with positivity rates of 7.1% when compared with 58.3% for β-D-glucan [45]. Although a detailed analysis of the *Fusarium* cell wall composition is lacking, the low positivity rate of the GM assay in IF suggests that *Fusarium* contains less galactomannan in its cell wall compared to *Aspergillus*. In contrast, β-D-glucan, a more conserved and widely distributed fungal cell wall component, exhibited more readily detectable in the bloodstream, likely due to its higher abundance and consistent presence across fungal species. Finally, DNA-based diagnostic methods, such as qPCR, have demonstrated high specificity for *Fusarium*; fungal elements were detectable for up to 18 days prior to confirmation by blood culture or biopsy (median detection time: six days) [45]. Casalini et al. found that qPCR detected circulating *Fusarium* DNA with no false positives among patients with other invasive fungal diseases (n = 12) or IFD-free controls (n = 40) [3].

## 3. Intrinsic Azole Resistance of Fusarium

### 3.1. Resistance Evolution

Azole resistance in clinical *Fusarium* isolates may arise from prolonged antifungal therapy regimens for invasive and chronic infections (Figure 1) as well as from the hospital environment [8]. While both agricultural and medical triazoles share the same molecular target, it is plausible to assume that clinical resistance could be exacerbated by the extensive use of azole fungicides in agriculture. Azoles are widely applied in agriculture to protect food crops from fungal pathogens such as *Fusarium*. Their widespread use on every continent is driven by affordability, broad-spectrum antifungal activity, and long-lasting stability, making them a preferred choice for crop protection [46,47,48]. Since the first agricultural azole, imazalil, was introduced in 1973—two decades before fluconazole—over 25 azoles have been developed for agricultural applications, whereas fewer than 10 clinical triazoles have been introduced [49,50].

The cross-kingdom pathogenicity of *Fusarium*, affecting both humans and plants, results in significant exposure to azole antifungals from agricultural soil, antifungal-treated patients, and the hospital environment. In hospitals, *Fusarium* species have been detected in the air of patient rooms, water tanks, drains, showerheads, and aerosolized water following shower use [8]. In the agricultural field, the primary pathogenic species within the FSSC and the FOSC complexes are commonly found in soil, plants, and water, where they can also cause waterborne diseases in animals and crops [51,52,53,54]. Additionally, *Fusarium* species have been detected in wastewater treatment plant effluents, which can disseminate into rivers, further contributing to environmental exposure [55]. However, there is currently a lack of experimental evidence to support the connection between azole resistance in these two environmental niches.

The treatment of invasive fusariosis (IF) and localized fusariosis remains challenging, as *Fusarium* exhibits significant resistance to antifungal agents. In vitro studies have demonstrated that *Fusarium* isolates show poor sensitivity not only to first-generation triazoles such as fluconazole and itraconazole but also to amphotericin B (AmB) and the second generation of triazoles, including voriconazole (VOR), posaconazole (POS), and isavuconazole (ISA) [56,57]. Despite this resistance, voriconazole (VOR) and amphotericin B (AmB) remain the primary treatment options for invasive fusariosis (IF), with VOR frequently used prophylactically in clinical practice [58].

Our own studies, which include 29 clinical isolates provided by the CDC and The University of Texas at San Antonio, indicate that 85% of these strains are resistant to VOR. Currently, we are evaluating over 100 strains isolated from nail infections for resistance to azoles, amphotericin B, caspofungin, and other antifungals in vitro. Preliminary data suggest that cross-azole resistance and multidrug resistance (MDR) are prevalent among clinical isolates in the United States (manuscript in preparation).

### 3.2. Molecular Basis of Fusarium Azole Resistance

Three major azole resistance mechanisms have been extensively studied in *C. albicans* and the mold *A. fumigatus*: (i) mutations in the azole target genes (*CYP51*), which account for 50–80% of *Aspergillus* resistance cases; (ii) overexpression of the target gene; and (iii) upregulation of efflux pumps [59,60]. In *Aspergillus*, the first two mechanisms are dominant, with mutations in *CYP51A* playing a key role [57]. A well-characterized resistance-associated alteration in *Aspergillus* involves a leucine-to-histidine substitution at codon 98 (L98H), accompanied by a 34-base pair tandem repeat (TR34/L98H) insertion in the *CYP51A* promoter. This mutation, frequently identified in azole-naïve patients with aspergilloma and lung disease, is considered a hallmark of environmentally acquired resistance [61,62]. The tandem repeat in the promoter contributes to the overexpression of *CYP51A*.

In contrast, the molecular mechanisms of azole resistance in *Fusarium*, particularly those involving *CYP51A*, remain less well characterized in clinical isolates. While amino acid substitutions from point mutations alter target interactions with azoles in *Aspergillus*, this mechanism does not appear to be relevant to resistance in field-isolated *F. graminearum* and *F. asiaticum* [63,64], both of which belong to the *Fusarium* graminearum species complex (FGSC). However, laboratory-induced metconazole adaptation resulted in mutations associated with different expression patterns in *F. graminearum CYP51* genes [65]. *Fusarium* possesses several genetic features that likely contribute to its intrinsic resistance to azoles and even amphotericin B (AmB) [57,66]. Notably, this intrinsic resistance has been reported in clinical isolates collected both before and after 1990 [57], when fluconazole was first introduced in clinical settings. This suggests that the evolutionary path of azole resistance in *Fusarium* differs from that of *Aspergillus*.

One possible explanation for *Fusarium*’s intrinsic resistance is its high genomic plasticity, which may be facilitated by horizontal gene transfer [67]. Furthermore, recent discoveries of aneuploidy-driven resistance in drug-resistant *Cryptococcus neoformans* [68,69] and RNA interference (RNAi)-mediated resistance in *Mucor* spp. [70,71] suggest that *Fusarium* may employ similar mechanisms, making its antifungal resistance more complex than previously anticipated. These findings highlight the need for further investigation into the genetic and epigenetic basis of *Fusarium* drug resistance.

#### 3.2.1. Point Mutations in CYP51 Paralogs

Regarding the classic point mutation mechanism, *Fusarium* possesses two key genetic advantages that may contribute to its azole resistance. First, it has three paralogs of the azole target gene (*CYP51A*, *CYP51B*, and *CYP51C*), whereas *C. albicans* has only one (*ERG11*), and *Aspergillus* has two (*CYP51A* and *CYP51B*). Second, *Fusarium* has those known missense mutations within their three CYP51 paralogs (Table 1). In *Aspergillus*, three hotspot regions for point mutations in CYP51A have been identified with azole resistance (Figure 2) [72]. These mutations can occur independently (e.g., any or all of G54I, M220L, G138C, and Y121F) and are primarily observed in aspergilloma patients with lung cavities. Some of these hotspot point mutations occur in combination with tandem repeat promoter alterations, such as TR34/L98H and TR46/Y121F/T289A, which are common in environmentally resistant *A. fumigatus* strains. In silico modeling of *A. fumigatus CYP51A* has revealed that substitutions at residues L98, M220, or Y431 reduce azole binding affinity, while mutations at residues L98 or G432 decrease protein stability, potentially diminishing the antifungal effects of azoles [73].

BLAST analysis of randomly selected *Fusarium* genomes identified the M220L mutation in CYP51A of clinically relevant *Fusarium* species, along with six of twelve known *Aspergillus CYP51A* SNPs distributed across *CYP51A*, *CYP51B*, and *CYP51C* (Table 1). Four reported CYP51C missense mutations, including S240A—linked to voriconazole resistance in *Aspergillus flavus* [75]—had corresponding but distinct amino acid substitutions in *Fusarium* CYP51 paralogs. The impact of these SNPs on *Fusarium* azole resistance remains largely unexplored, particularly for CYP51C mutations located in non-conserved regions of *Fusarium* CYP51 paralogs. Nevertheless, the presence of preexisting *CYP51A*-associated SNPs across all three *Fusarium CYP51* paralogs may contribute to its inherent molecular basis for azole resistance, as CYP51A plays a more significant role in drug response (see the next section). While mutations in hotspot 1 and hotspot 2 of CYP51A confer itraconazole resistance in *Aspergillus* [72], alterations such as L98H, M220I (L), and T289A could contribute to the intrinsic insensitivity of clinical *Fusarium* isolates to itraconazole, which was not specifically designed to target *Fusarium* CYP51 paralogs.

Independent research by the authors (now in press) also confirms that it is not even possible to determine an MIC for itraconazole, so pervasive is the resistance of clinical strains to this drug. In field-isolated *F. graminearum* from China, point mutations such as S28L, S256A, and V307A in CYP51C were observed in hexaconazole-resistant strains [76]. By contrast, laboratory-induced metconazole-resistant *F. graminearum* harbored mutations such as D243N, E103Q/V157L, and G443S in CYP51A [65], which differed from the CYP51A variants identified in hexaconazole-resistant strains [76]. Without phenotypic confirmation of drug susceptibility, it remains inconclusive whether these reported mutations in these plant-pathogenic *Fusarium* species are influenced by the specific azole used in the study or the induction conditions. Notably, while all CYP51C mutations fall outside conserved amino acid regions [76], CYP51A mutations occur within conserved regions or putative substrate-recognition sites (SRS) [77] in all three CYP51 paralogs across different *Fusarium* species. Therefore, verifying the association between these CYP51A SNPs or other distinct SNPs linked to clinical azole-resistant *Fusarium* species is of particular interest. The widespread occurrence of these SNPs in available genomic data, along with corresponding azole susceptibility profiles, warrants further large-scale investigation into their potential role in clinical azole resistance.

#### 3.2.2. Overexpression of *CYP51* Genes and Exclusion Drug by Efflux

Overexpression of the target gene enhances substrate binding, allowing competition with the azole drug. This enables the fungus to sustain ergosterol synthesis and survive to a certain extent [78,79]. Data on the overexpression of *CYP51* genes in *Fusarium* are limited. Notably, all three *Fusarium* paralogs are transcriptionally active in both the mycelial and conidial phases [80] and can be up-regulated under voriconazole (VOR) treatment [9]. A study by James et al. examined the expression responses of these paralog genes in *F. solani*, showing that *CYP51B* is highly expressed under normal growth conditions without drug pressure [9]. In contrast, *CYP51A* was upregulated by several thousand-fold upon exposure to 1 µg/mL VOR. The same level of VOR led to a moderate increase of ~20-fold in *CYP51B*, and an increase of 3–6-fold in *CYP51C* expression. Apparently, *CYP51A* functions as an inducible responder to azole pressure, playing a crucial role in adaptive azole resistance. In agreement with the constitutive expression of *CYP51B*, deletion mutants of *CYP51A* (*Δcyp51A*) or *CYP51C* (*Δcyp51C*), but not *CYP51B* (*Δcyp51B*), in *Fusarium graminearum* were found to be more susceptible to agricultural azole inhibitors [77]. However, the ergosterol content in *Δcyp51A*, *Δcyp51B*, or *Δcyp51C* mutants did not differ from that of the wild-type strain, suggesting functional redundancy among these paralogs. This redundancy may also be reflected in laboratory-induced metconazole mutations in *F. graminearum*. Beyond the biological fitness penalty, the D243N mutation in CYP51A was associated with the overexpression of all three *CYP51* genes, whereas E103Q/V157L led to the overexpression of *CYP51A* and *CYP51B*, and G443S was linked to *CYP51A* only. These findings suggest that the impact of single SNPs in any one paralog on azole resistance may be limited.

Although the efflux mechanism has been considered to play a minor role in azole resistance in clinical mold isolates, some studies on field-isolated *F. graminearum* suggest otherwise. The activation of ABC transporters, such as the plasma membrane-localized H^+^ antiporter FgQdr2, significantly reduces *F. graminearum*’s sensitivity to azoles [81,82]. However, the role of efflux pumps and their regulatory mechanisms in azole resistance remains poorly understood.

#### 3.2.3. Promoter Mutations in *CYP51A* and Their Potential Impact on Signal Transduction Pathways

Interestingly, a recent study on 25 *F. solani* species complex (FSSC) isolates from Malaysia identified a 23 bp *CYP51A* promoter deletion (DEL) in nine isolates with voriconazole MICs exceeding 32 µg/mL [9]. These isolates originated from diverse clinical and environmental sources. Although the authors proposed this deletion as a potential marker for voriconazole resistance in FSSC isolates, its direct role remains uncertain, as it only led to a modest *CYP51A* upregulation (1.3- to 7.5-fold). This 23bpDEL mutation is located 120 bp upstream of a putative 16 bp sterol regulatory (SR) *cis*-element, which has been identified as a binding site for the zinc-cluster transcription factor FgSR in *Fusarium graminearum* [83]. Among 116 genes with *FgSR* binding sites, 20 encode ERG proteins involved in ergosterol biosynthesis, including *CYP51A*, *CYP51B*, and *CYP51C*. Notably, deletion of the *FgSR* gene resulted in an 85% reduction in ergosterol levels and increased susceptibility to azoles and DNA synthesis inhibitors. These findings highlight a more pronounced role of *FgSR* in regulating ergosterol biosynthesis and the DNA damage response under fungicidal stress, potentially overriding the resistance influence of *CYP51* paralogs.

Notably, FgSR orthologs are found only in two other ascomycete fungal classes, *Sordariomycetes* and *Leotiomycetes*. The authors demonstrated that *FgSR* activation occurs through phosphorylation in the nucleus upon ergosterol depletion, mediated by the Hog1 MAPK pathway. This regulatory mechanism differs from the canonical pathways governing ergosterol biosynthesis, such as the nuclear and ER translocation of Upc2 in *Candida* and SREBP (sterol regulatory element-binding protein) in *Aspergillus* [84,85]. Further investigation is needed to determine how promoter mutations in *Fusarium CYP51* paralogs influence *FgSR* binding activity, particularly in the inducible *CYP51A*.

Understanding the role of this Hog1-MAPK signaling pathway in azole resistance in *Fusarium* is particularly relevant given the differences in azole susceptibility between yeast pathogens and *Fusarium* mutants of this pathway. In *Candida*, the Hog1 MAPK pathway is upregulated in response to stress resistance, including oxidative, osmotic, and drug-induced stresses. This pathway plays a critical role in regulating cell morphology, aggregation, and virulence. As a result, the Hog1 MAPK pathway has been suggested as a potential target for antifungals, especially in combating emerging drug-resistant pathogens [86,87,88]. In *Cryptococcus*, deletion of HOG1 confers increased susceptibility to azoles [89]. Both yeast responses contrast with the loss-of-function mutations in HOG1 and related MAPK components in *F. graminearum*, which increase azole susceptibility [86]. This raises an intriguing question: how do host-derived reactive oxygen species (ROS) during infection regulate the Hog1-FgSR pathway? Further studies are needed to clarify the impact of this pathway on ergosterol levels and azole susceptibility in *Fusarium* infection.

In summary, although *Fusarium* and *Aspergillus* are both mold pathogens with trans-kingdom infection potential, the currently characterized azole resistance mechanisms in *Aspergillus* do not necessarily apply to *Fusarium*. Growing evidence highlights differences between these two genera in ergosterol regulation and the impact of known resistance-associated SNPs. These observations emphasize the need for further functional studies to determine the precise contributions of *Fusarium CYP51* paralogs to azole resistance and to explore additional resistance mechanisms, including epigenetic regulation, genomic adaptation strategies, and the signal transduction pathways.

## 4. *Fusarium* Biofilms

Fungal biofilms are a critical virulence factor, providing protection against host (human/plant) defenses and environmental conditions while increasing drug resistance. In *Fusarium*, biofilm formation is particularly relevant in keratitis and onychomycosis [34,90] the most common clinical manifestations of fusariosis. Biofilms have been demonstrated in corneal infections, contact lenses, and nail infections, where they contribute to persistence and therapeutic challenges.

### 4.1. Biofilm Formation in Fusarium Infections in Humans

*Fusarium* keratitis is frequently observed in contact lens users, where fungal cells attach to the lens case, secrete proteins, and establish a biofilm [91]. Beyond colonizing living surfaces such as the cornea and nails, biofilm-like structures are also found on non-living medical devices, including contact lenses, prosthetic heart valves, coronary stents, and prosthetic joints. Biofilm development follows a five-stage process: (i) reversible attachment of floating fungal cells to the lens surface; (ii) irreversible attachment and colonization; (iii) expansion of fungal layers; (iv) formation of a protective extracellular matrix (ECM); and (v) detachment and dispersal of fungal cells into the surrounding environment. Scratches or micro-injuries on living surfaces, such as the corneal or nail epithelium, facilitate fungal penetration and biofilm establishment. Additionally, contact lenses reduce the antimicrobial efficacy of the tear film, further promoting keratitis infection.

### 4.2. In Vivo and Ex Vivo Models of Fusarium Biofilm Formation

Several in vivo models have been developed to study *Fusarium* biofilms in keratitis using immunocompromised rabbits and mice [92]. In immunosuppressed mice, co-infection of *Fusarium falciforme* with *Staphylococcus aureus* led to biofilm formation in the eye [93]. Histopathological evaluation of enucleated eye sections revealed an extracellular matrix (biofilm) on the surface of both organisms. In a cyclophosphamide-treated BALB/c mouse model, *F. solani* infection alone resulted in biofilm-like fungal masses in the cornea within 4–8 days, accompanied by fungal infiltration and neutrophil accumulation in the anterior chamber [94]. Similarly, in a rabbit model pretreated with corticosteroids and antibiotics, a biofilm-like mass formed one week after interlamellar injection of *F. solani* [95].

The role of biofilms in *Fusarium* onychomycosis has been investigated using an ex vivo infection model with *F. oxysporum* isolates from an onychomycosis patient [34]. Microscopic analysis revealed conidial transition to hyphae and biofilm formation by day six. ECM analysis detected lipids, proteins, polysaccharides, extracellular DNA (eDNA), and extracellular RNA (eRNA), consistent with ECM components observed in in vitro biofilms. These findings strongly suggest that biofilms in vivo may contribute to its antifungal resistance and recurrence in *Fusarium* nail infections [33].

### 4.3. In Vitro Studies on Fusarium Biofilms

In vitro biofilm formation follows the same progression as in vivo and ex vivo models, involving adherence, germination, hyphal development, maturation, and detachment [90]. Biofilm ECM in vitro also consists of carbohydrates, proteins, lipids, and eDNA. Optimal biofilm development occurred with an inoculum of 1 × 10^6^
*F. falciforme* conidia in a 96-well plate using RPMI medium at 28 °C for 96 h [96] in an in vitro biofilm testing system. Proteomic analysis identified 19 overexpressed proteins in biofilms, including transketolase, enolase, phosphoglycerate kinase, and ATP-citrate synthase, highlighting that glycolysis and gluconeogenesis play significant roles in biofilm formation [96].

The crucial role of these metabolic pathways aligns with our phenotypic observations [97]. In that study, clinical isolates from four major *Fusarium* species complexes (*F. solani*, *F. fujikuroi*, *F. oxysporum*, and *F. incarnatum-equiseti*) were analyzed for biofilm composition and antifungal response. Notably, these clinical *Fusarium* species, like plant pathogens, exhibit poor thermotolerance and anaerobic growth, which explains their preference for superficial infections in the eyes and nails, where oxygen and lower temperatures are available. Despite variability among species, glucose promotes biofilm formation more effectively than other less favorable sugars or glycerol. Among the four *Fusarium* species complexes, *F. oxysporum* demonstrates greater nutritional flexibility during biofilm formation. Our second key finding is that azole-resistant *F. solani* strains form more robust biofilms compared to azole-susceptible strains when exposed to antifungals, including voriconazole (VOR), amphotericin B (AmB), and 5-flucytosine (5-FC). This suggests that the metabolic flexibility inherent in *Fusarium* and the drug stress adaptation mechanisms, which likely develop pre-clinically, enhance the survival of these strains and facilitate biofilm formation within the host.

### 4.4. Biofilm Composition Analysis and Treatment Implications

Over time, ECM composition shifts from lipid-rich in early stages to polysaccharide- and protein-dominant in late stages in our recent study with clinical *Fusarium* isolates [97]. We found that VOR demonstrated greater efficacy than AmB in inhibiting *Fusarium* biofilm formation. Our findings also highlight the potential of antifungal combination therapies to disrupt *Fusarium* biofilms and improve treatment outcomes by targeting the distinct composition of the biofilm. For example, lipids in the ECM during the early stages of biofilm formation could serve as a potential intervention target. In an in vivo biofilm model, lipids in the nail biofilm extracellular matrix (ECM) were found to peak at 7 days post-infection [34]. These secreted lipids, detected using FTIR-ATR spectroscopy, exhibit characteristic functional groups, including C=O (associated with triglycerides and phospholipids), C-N (found in fatty acid amides, which function as signaling molecules), and CH_2_/CH_3_ structures. However, the specific lipid classes, such as phospholipids or sterol lipids, within the *Fusarium* biofilm remain to be fully characterized.

The amount of ergosterol in the ECM of *C. albicans* varies with biofilm development [98,99]. Consistent with the early appearance of lipids observed in clinical *Fusarium* isolates, ergosterol content in *C. albicans* is highest during the early stages of biofilm formation and is reduced by approximately 50% in mature biofilms [98]. Given that voriconazole targets sterol lipids and amphotericin B interferes with phospholipid structures, the superior anti-biofilm efficacy of VOR [97] suggests a crucial role for sterol lipids in biofilm development. Interestingly, biofilm compositions from *Fusarium graminearum*, a plant pathogen, exhibit distinct characteristics, including a lack of lipids [100]. When testing other biofilm characteristics of this plant pathogen, it was observed that under anaerobic conditions, short bulbous hyphae were surrounded by ECM in the absence of electron acceptors. The addition of electron acceptors restored filamentous growth. This suggests that elevated reactive oxygen species (ROS), typically produced by the host response during infection, may trigger hyphal development during biofilm formation.

## 5. Summary and Prospect

*Fusarium* spp. are opportunistic mold pathogens that cause a range of infections, including keratitis and onychomycosis. Biofilm formation plays a role in the persistence of these infections, providing protection against environmental stressors and antifungal treatment. While biofilms enhance resistance to some antifungal agents, emerging evidence suggests that *Fusarium* azole resistance is primarily driven by distinct molecular mechanisms rather than biofilm formation alone. In contrast to *Aspergillus*, *Fusarium* has unique ergosterol biosynthesis regulation, with multiple CYP51 paralogs playing a key role in its resistance. Current evidence suggests that the severe azole resistance observed in clinical *Fusarium* isolates may result from multiple mechanisms, which differ from the single nucleotide polymorphism (SNP)-driven mechanisms seen in *Aspergillus*. Regardless of the specific genomic adaptations, other resistance mechanisms—such as epigenetic modifications and signal transduction pathways—may also influence *Fusarium*’s azole resistance. Gaining a deeper understanding of these *Fusarium*-specific mechanisms is crucial for developing targeted antifungal strategies to improve treatment outcomes.

## Figures and Tables

**Figure 1 jof-11-00294-f001:**
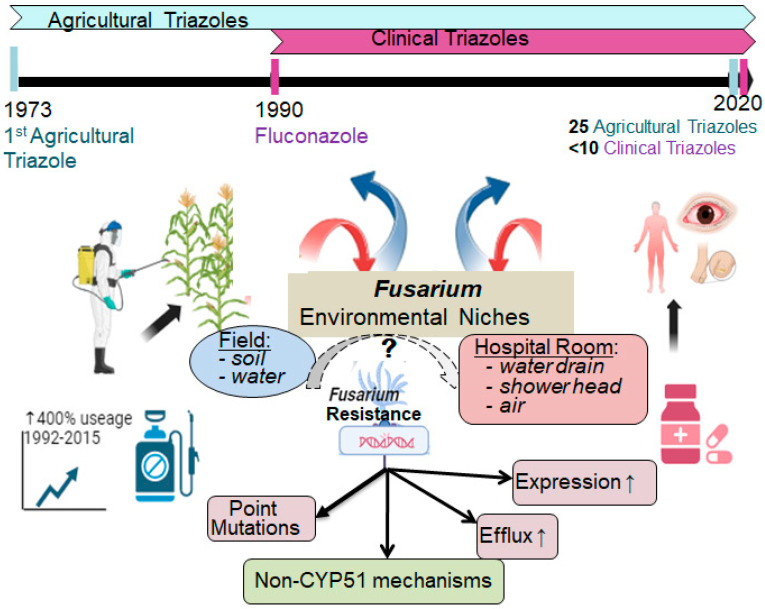
Triazole resistance in *Fusarium* may have evolved in two environmental niches: (i) soil, crops, and water exposed to agricultural azole use and (ii) hospital environments where *Fusarium* is shed from patients. *Fusarium* species have been detected in hospital environments, including air in patient rooms, water tanks, showerheads, and drains. Agricultural triazoles have a broader range and a longer history of use compared to clinical triazoles. However, there is currently no genetic evidence to distinguish the origin of triazole resistance in *Fusarium* (dashed arrow), as seen with the L98H/TR34 mutations in *A. fumigatus*, which indicate an environmental origin. Resistance may also emerge during prolonged antifungal treatment, particularly in superficial fusariosis—such as keratitis and onychomycosis—which often require extended therapy, potentially driving resistance development in clinical settings. Resistance mechanisms may involve *CYP51* paralog-associated mechanisms (magenta boxes) and non-*CYP51* mechanisms (green box).

**Figure 2 jof-11-00294-f002:**
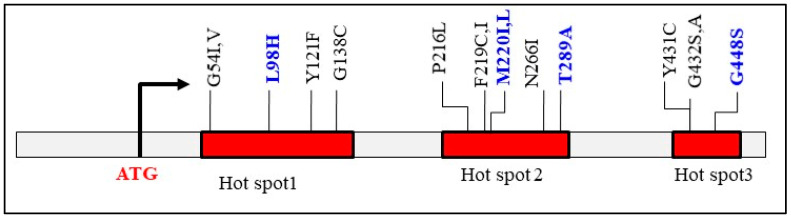
Known missense mutations in *A. fumigatus CYP51A* that confer azole resistance are concentrated in three hotspot regions (Ref. [72]). Mutations L98H and G138A in hotspot 1, M220I (L), N266I, and T289A in hotspot 2, as well as G448S in hotspot 3, are also present in orthologous CYP51 proteins of wild-type *Fusarium* spp.

**Table 1 jof-11-00294-t001:** Corresponding amino acid alterations in three *Fusarium* species associated with known azole resistance missense mutations in *A. fumigatus* and *A. flavus*.

Residues * of Missense Mutations in *A. fumigatus* CYP 51 Proteins	*F. solani* CYP51			*F. oxysporum* CYP51			*F. fujikuroi* CYP51		
	A	B	C	A	B	C	A	B	C
*A. fumigatus* CYP51A									
G54I, Y121F	-	-	-	-	-	-	-	-	-
L98H	-	H	-	-	H	-	-	H	-
G138C	-	-	A	-	-	A	-	A	-
P216L, F219C	-	-	-	-	-		-	-	-
M220I	L	I	-	L	V	-	L	-	V
N266I	-	T	H	-	T	H	-	H	T
T289A	-	A	A	-	A	A	-	A	A
Y431C, G432S	-	-	-	-	-	-	-	-	-
G448S	-	-	A	-	-	S	-	-	S
*A. fumigatus* CYP51B (Ref. [74])									
G457S	-	-	-	-	-	-	-	-	-
*A. flavus* CYP51C (Ref. [75])									
M54T	L	-	I	L	-	I	L	-	I
S240A **	D	K	G	E	K	E	E	K	E
P419T	V	-	D	-	-	E	-	-	E
N423D	A	G	A	A	G	K	V	A	V

*: Amino acid and position in *A. fumigatus* CYP51A and CYP51B; **: S240A missense mutation in CYP51C confers voriconazole resistance in *A. flavus*; -: no change observed; mutations in hotspot 1 hotspot 2, and hotspot 3 are marked in yellow, green, and grey backgrounds, respectively; *F. solani* CYP51A (QGR26263.1), CYP51B (QGR26271.1), and CYP51C (QGR26274.1); *F. oxysporum* CYP51A (EXK90156.1), CYP51B (EXK89719.1), and CYP51C (RKK08703.1); *F. fujikuroi* CYP51 (CCT73654.1), CYP51B (CCT62283.1), and CYP51C (KAF4444228.1).

## Data Availability

No new data were created or analyzed in this study. Data sharing is not applicable to this article.

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
