# Peer review of "The Dual Pathogen Fusarium: Diseases, Incidence, Azole Resistance, and Biofilms"

_jof, 2025, doi:10.3390/jof11040294_

Round 1
Reviewer 1 Report
The is an interesting and well-writen review covering different aspects of Fusarium infections and in particular its resistance to antifungals. I have just a few minor comments.
l. 39: "must be due" is very strong. Next sentence: delete "of"
l. 46: space after Fusarium
l. 48: Is Ref 10 correct?
l. 66: can you provide a reference for this?
l. 72: please explain GAFFI, 2022 in parentheses?
l. 74: if these umbers are estimates, I would adjust them
l. 88-90: the logic behind this sentence is difficult to grasp
l. 108: is the structure of the nail another factor effecting the distribution of antifungals?
l. 119: "such as" the construction of this sentence can be improved
l. 158. replace of by as?
l. 160: to my nowledge azoles are not extensively used in veterinary medicine
l. 167: this is true, but exposure in the soil is also quite important
Figure 1 useage, some words start with cpital letters, other not
l. 197: "is contributed"
Table 1: several spaces can be introduced after species names
l. 241: "along with correspsonding azole suceptibility profiles" the problem seems to be that these profiles are rare. The authors may rewrite hs sentence, which may be better positioned at the end of ths paragraph.
l. 258: are these genes "transcriptionally active" in conidia or are mRNAs stored in these spores?
l. 310: "contrast contrasts"
l. 331: fungal cells?, is protein secretion important?
l. 360/61: please rephrase this sentence
l. 363: The mentioned optimal number of conidia. relevance? volume? incubation time?
l. 402: space after [95]
Author Response
Major comments: The is an interesting and well-written review covering different aspects of Fusarium infections and in particular its resistance to antifungals. I have just a few minor comments.
Response: We sincerely appreciate your thoughtful and constructive feedback on our manuscript.
Minor comments:
Comment 1: 39: "must be due" is very strong. Next sentence: delete "of"
Response: Thank you to point these out. We have replaced “must” with “may”, and deleted “of”. in line 40
Comment 2: 46: space after Fusarium
Response: A “space” is added
Comment 3: l. 48: Is Ref 10 correct?
Response: We have replaced the old reference with a more appropriate one, and added one more reference as [8] in line 49
Comment 4. l. 66: can you provide a reference for this?
Response: The statement is the overall summary of risk factors commonly found in invasive and superficial fusariosis. References [8-9,11] are added in line 68.
Comments 5. l. 72: please explain GAFFI, 2022 in parentheses?
Response: We have added the full name of GAFFI for Global Action For Fungal Infections in line 74
Comments 6. l. 74: if these numbers are estimates, I would adjust them
Response: We agree with you. As these numbers are estimates, we rephrased the text (highlight in red) in lines of 75-76
Comments 7. l. 88-90: the logic behind this sentence is difficult to grasp
Response: Thanks for pointing this out, we have rephrased the sentence to highlight the need for treatment strategy to reduce the chronicity and potential latency in lines of 90-92.
Comment8. l. 108: is the structure of the nail another factor effecting the distribution of antifungals?
Response: Yes. This is also one of reason to cure the onychomycosis (no blood circulation). We added nail structure as a factor as well, in line 111.
Comment 9 l. 119: "such as" the construction of this sentence can be improved
Response: The sentence has been revised as you suggested, see lines of 122-124
Comment 10 l. 158. replace of by as?
Response: changed to "as", see line 164
Comment 11. l. 160: to my knowledge azoles are not extensively used in veterinary medicine
Response: Thanks for reminding, we deleted veterinary medicine in the sentence.
Comment 12. l. 167: this is true, but exposure in the soil is also quite important
Response: Agree, we have rephrased the sentence to reflect the soil in agricultural and hospital environment in lines of 173-175, and Figure 1 legend
Comment 13. Figure 1 useage, some words start with cpital letters, other not
Response: Thanks for pointing it out. We have redrawn the Figure 1 and capitalized all phrases when needed.
Comment 14. l. 197: "is contributed"
Response: changed to "contributes"
Comment 15. Table 1: several spaces can be introduced after species names
Response: Thanks for pointing it out. We have added the spaces after species names.
Comment 16. l. 241: "along with corresponding azole susceptibility profiles" the problem seems to be that these profiles are rare. The authors may rewrite their sentence, which may be better positioned at the end of this paragraph.
Response: The sentence has been moved in the end of paragraph and entire paragraph (lines of 261-277) was revised.
Comment 17. l. 258: are these genes "transcriptionally active" in conidia or are mRNAs stored in these spores?
Response: Transcriptionally active remains (line 283). Since all three genes are transcribed when grew at the normal and drug conditions.
Comment 18. l. 310: "contrast contrasts"
Response: We deleted “contrasts”
Comment 19 l. 331: fungal cells?, is protein secretion important?
Response: We have replaced “microbial” to “fungal” (line 362). The protein secretion is important regarding the biofilm composition.
Comment 20 l. 360/61: please rephrase this sentence
Response: We have revised this sentence in lines 393-394
Comment 21. l. 363: The mentioned optimal number of conidia. relevance? volume? incubation time?
Response: Thanks for pointing this out. We have rephrased the sentence to reflect the relevance duration and temperature, see lines of 396-397.
Comment 22. l. 402: space after [95]
Response: A space added, in line 403 after [98]
Reviewer 2 Report
The authors of this review present information on the clinical types and incidence of human fusariosis and explore the mechanisms involved in the intrinsic azole resistance of Fusarium. The manuscript in general is well written and structured. However, it has certain shortcomings that need to be addressed with respect to azole resistance and Fusarium biofilms. For the azole resistance, the authors attempt to make a connection between agricultural and clinical Fusarium azole resistance, when they acknowledge the lack of markers for differentiating hospital from agricultural Fusarium strains. They explore the potential genetic basis of Fusarium azole resistance based on putative amino acid changes generated by computer-assisted sequence homology between Fusarium species complexes and Aspergillus fumigatus; such an approach, however, does not support their hypothesis that azole resistance of clinical Fusarium strains may come from agricultural azole resistant strains. For the biofilms, the authors need to present and discuss in vitro/in vivo studies of Fusarium biofilms against azole antifungals and do a comparative analysis in order to draw relevant conclusions on azole resistance.
Specific Comments
- Although the authors agree that azole resistance is not linked to agricultural azole use (pg 1, lines 36 and 37 and lines 39-41), as it is documented for Aspergillus, Figure 1 depicts and Figure 1 legend states that triazole resistance is driven by agricultural azole use.
The data that support the hypothesis that azole resistant A. fumigatus strains are acquired from field-based environmental sources come not only from patients who were never been treated with azole antifungal drugs, but also from the fact that azole resistant A. fumigatus clinical strains were found in many environmental niches including flowerbeds, compost, leaves, plant seeds, soil samples of tea gardens or paddy fields (cited ref 7, doi: 10.1371/journal.ppat.1003633). Additionally, the article cited on invasive fusariosis (ref 8, doi: 10.1128/cmr.00159-22) reports on cases of fusariosis due to Fusarium species that were found in the hospital environmental sources such as in the air of hospital rooms, in the water tanks, shower heads, drains, or aerosolization of Fusarium species occurring after running the showers; they were not found in agricultural areas (line 33). The cited articles that correspond to references 51-55 are Fusarium species pathogenic to wheat or plants, with no evidence of being human pathogens as well (pg 4, lines 168-172). The above cited articles rather show the lack of connection between clinical and agricultural azole resistance.
To preserve the scientific accuracy, please be clear about the above point. Figure 1 must be redrawn to depict precise information and Figure 1 legend should be rephrased in order to agree with the main text of the manuscript.
- For the sake of clarity and completeness of the review, missense mutations found in the homologous fumigatus CYP51B should also be reported in Table 1. In fact, it would make better sense to the reader if the data presented in Table 1 were shown in terms of direct gene-by-gene comparison. Since both Aspergillus and Fusarium contain CYP51A and CYP51B genes, it is recommended, to contrast the relevant deduced missense mutations found in A. fumigatus CYP51A vs F. solani, F. oxysporum and F. fujikuroi CYP51A genes and between A. fumigatus CYP51B vs F. solani, F. oxysporum and F. fujikuroi CYP51B genes, respectively. Likewise it is recommended, for missense mutations found in CYP51C of Aspergillus flavus that have been described and associated with azole resistance to be compared with relevant positions in CYP51C of F. solani, F. oxysporum and F. fujikuroi. Computer-assisted 3D structural motif analyses will reveal whether or not there are amino acid changes in the relevant positions that are reported to be associated with azole resistance or could potentially play a role in azole resistance due to conformational changes reported and/or demonstrated. The current data presented in Table 1, are rather confusing as presented. Please change/modify.
- In the legend of Figure 2, please write out the 5 missense mutations identified in Fusarium orthologs because symbols and colors are not discernible in the presented scheme (Figure 2).
- The fumigatus missense mutations shown in Table 1 do not coincide with those shown in Figure 2. Please review and correct.
- The authors should be consistent with the representation of missense mutations: In the 3.2.1 section, Table 1 and Figure 2, missense mutations are shown using different symbols (L98H and L98; Y121F and Y121 etc).
- Is the promoter alteration T298A (3.2.1. subsection) or T289A (Figure 2, Table 1)? Please correct the typo.
- Please explain/define the meaning of the term “mutation codes” (pg 6, line221).
- The sentence, “However, amino acid substitutions resulting from point mutations can alter target interactions with azoles and even enhance the expression of target genes, both of which are recognized as common resistance mechanisms in field-isolated Fusarium” (pg 5-6, lines 200-203), does not correspond to results obtained in the cited article, ref. 63. The article by Yin et al (DOI: 1094/PHYTO-99-5-0487) reports that “Analysis of deduced amino acid sequence of cyp51A and cyp51B suggested that no mutations were associated with DMI resistance. Real-time PCR analysis showed that the DMI resistance was not related to the expression of cyp51A and cyp51B in F. asiaticum and F. graminearum…..” Therefore, the azole resistance mechanism involved was unrelated to enhanced expression of target genes, as it is reported in the current manuscript (pg 5-6, lines 200-203). Please correct.
- The subsection 4 on Fusarium biofilms and azole resistance (pg 9-10, lines 323-409) primarily describes the life cycle of biofilms, in vivo models of Fusarium biofilm forming capacity and in vitro studies of biofilm composition. The above information should be shortened to one paragraph of 15 lines at the most, in order to give pertinent information on in vitro/in vivo studies regarding the antifungal activity of azoles against Fusarium biofilms.
Author Response
Major comments
The authors of this review present information on the clinical types and incidence of human fusariosis and explore the mechanisms involved in the intrinsic azole resistance of Fusarium. The manuscript in general is well written and structured. However, it has certain shortcomings that need to be addressed with respect to azole resistance and Fusarium biofilms. For the azole resistance, the authors attempt to make a connection between agricultural and clinical Fusarium azole resistance, when they acknowledge the lack of markers for differentiating hospital from agricultural Fusarium strains. They explore the potential genetic basis of Fusarium azole resistance based on putative amino acid changes generated by computer-assisted sequence homology between Fusarium species complexes and Aspergillus fumigatus; such an approach, however, does not support their hypothesis that azole resistance of clinical Fusarium strains may come from agricultural azole resistant strains. For the biofilms, the authors need to present and discuss in vitro/in vivo studies of Fusarium biofilms against azole antifungals and do a comparative analysis in order to draw relevant conclusions on azole resistance.
Response: We sincerely appreciate your thoughtful and constructive feedback on our manuscript. Your insights have been invaluable in refining our discussion on Fusarium azole resistance and biofilms. Regarding azole resistance, we recognized the need for a clearer connection between agricultural and clinical Fusarium strains. While our intent was to highlight potential "connection", we acknowledge that our approach does not discriminate hypothesis from directly established link. We have revised the MS to clarify our perspective and ensure that our conclusions are well supported. For the biofilm section, we appreciate your suggestion to incorporate a more comprehensive review of in vitro and in vivo studies in relation to azole resistance. However, due to the limited number of studies available on this topic, a thorough comparative analysis remains challenging. Nevertheless, we have refined our discussion to reflect the current state of knowledge while identifying gaps for future research. Your detailed feedback has significantly improved the quality of our manuscript, and we are truly grateful for your time and expertise.
Specific Comments
1. Although the authors agree that azole resistance is not linked to agricultural azole use (pg 1, lines 36 and 37 and lines 39-41), as it is documented for Aspergillus, Figure 1 depicts and Figure 1 legend states that triazole resistance is driven by agricultural azole use. The data that support the hypothesis that azole resistant A. fumigatus strains are acquired from field-based environmental sources come not only from patients who were never been treated with azole antifungal drugs, but also from the fact that azole resistant A. fumigatus clinical strains were found in many environmental niches including flowerbeds, compost, leaves, plant seeds, soil samples of tea gardens or paddy fields (cited ref 7, doi: 10.1371/journal.ppat.1003633). Additionally, the article cited on invasive fusariosis (ref 8, doi: 10.1128/cmr.00159-22) reports on cases of fusariosis due to Fusarium species that were found in the hospital environmental sources such as in the air of hospital rooms, in the water tanks, shower heads, drains, or aerosolization of Fusarium species occurring after running the showers; they were not found in agricultural areas (line 33). To preserve the scientific accuracy, please be clear about the above point. Figure 1 must be redrawn to depict precise information and Figure 1 legend should be rephrased in order to agree with the main text of the manuscript.
Response: Thank you for the constructive suggestion and careful review this article. To ensure scientific accuracy, we have revised Figure 1 to reflect the current lack of evidence directly connecting environmental resistance and clinical resistance.
2. The cited articles that correspond to references 51-55 are Fusarium species pathogenic to wheat or plants, with no evidence of being human pathogens as well (pg 4, lines 168-172). The above cited articles rather show the lack of connection between clinical and agricultural azole resistance.
Response: We completely agree. We have revised the second paragraph under Section 3.1, "Resistance Evolution," and highlighted the changes in red.
3. For the sake of clarity and completeness of the review, missense mutations found in the homologous fumigatusCYP51B should also be reported in Table 1. In fact, it would make better sense to the reader if the data presented in Table 1 were shown in terms of direct gene-by-gene comparison. Since both Aspergillus and Fusarium contain CYP51A and CYP51B genes, it is recommended, to contrast the relevant deduced missense mutations found in fumigatus CYP51A vs F. solani, F. oxysporum and F. fujikuroi CYP51A genes and between A. fumigatus CYP51B vs F. solani, F. oxysporum and F. fujikuroi CYP51B genes, respectively. Likewise it is recommended, for missense mutations found in CYP51C of Aspergillus flavus that have been described and associated with azole resistance to be compared with relevant positions in CYP51C of F. solani, F. oxysporum and F. fujikuroi. Computer-assisted 3D structural motif analyses will reveal whether or not there are amino acid changes in the relevant positions that are reported to be associated with azole resistance or could potentially play a role in azole resistance due to conformational changes reported and/or demonstrated. The current data presented in Table 1, are rather confusing as presented. Please change/modify.
Response: Thank you for the constructive suggestion. We have added the reported CYP51B mutations [Ref. 74] from A. fumigatus and CYP51C mutations from A. flavus [Ref. 75] to Table 1 and coordinate the text related to Table 1. We initially did not include missense mutations in CYP51B and CYP51C because CYP51B is constitutively expressed, while CYP51A is drug-inducible and plays a primary role in azole resistance mechanisms. All four mutations fall within non-conserved regions of CYP51 paralogs, distinguishing them from CYP51A missense mutations, which are located in conserved regions. Additionally, we suspect that the CYP51C mutations identified in A. flavus have a limited effect on Fusarium, as CYP51C plays a minor role in Fusarium's azole response. Our ongoing research continues to investigate drug resistance mechanisms, and we believe that, in addition to CYP51-mediated azole resistance, non-CYP51-mediated mechanisms may play a more significant role in Fusarium azole resistance.
4. In the legend of Figure 2, please write out the 5 missense mutations identified in Fusarium orthologs because symbols and colors are not discernible in the presented scheme (Figure 2).
Response: We have revised the legend for Figure 2 to indicate all the 5 missense mutations.
5. The fumigatus missense mutations shown in Table 1 do not coincide with those shown in Figure 2. Please review and correct.
Response: We have kept consistence of all the missense mutations between revised Table 1 and new Figure 2.
6. The authors should be consistent with the representation of missense mutations: In the 3.2.1 section, Table 1 and Figure 2, missense mutations are shown using different symbols (L98H and L98; Y121F and Y121 etc).
Response: Thank you for pointing it out. We have coordinated all the missense mutations in 3.2.1 section, Table 1 and Figure 2.
7. Is the promoter alteration T298A (3.2.1. subsection) or T289A (Figure 2, Table 1)? Please correct the typo.
Response: Thank you for pointing it out. We have corrected the error in the MS. It should be T289A.
8. Please explain/define the meaning of the term “mutation codes” (pg 6, line221).
Response: Thanks for suggestion. We have revised the “mutation codes” to “ missense mutations”, in line 236
9. The sentence, “However, amino acid substitutions resulting from point mutations can alter target interactions with azoles and even enhance the expression of target genes, both of which are recognized as common resistance mechanisms in field-isolated Fusarium” (pg 5-6, lines 200-203), does not correspond to results obtained in the cited article, ref. 63. The article by Yin et al (DOI: 1094/PHYTO-99-5-0487) reports that “Analysis of deduced amino acid sequence of cyp51A and cyp51B suggested that no mutations were associated with DMI resistance. Real-time PCR analysis showed that the DMI resistance was not related to the expression of cyp51A and cyp51B in asiaticum and F. graminearum…..” Therefore, the azole resistance mechanism involved was unrelated to enhanced expression of target genes, as it is reported in the current manuscript (pg 5-6, lines 200-203). Please correct.
Response: Thanks for your constructive comment. We have clarified this statement with “While amino acid substitutions from point mutations alter target interactions with azoles in Aspergillus, this mechanism does not appear to be relevant to resistance in field-isolated F. graminearum and F. asiaticum [63, 64], both of which belong to the Fusarium graminearum species complex (FGSC). However, laboratory-induced metconazole adaptation resulted in mutations associated with different expression patterns in F. graminearum CYP51 genes[65]. Reference 65 is the same of old Reference [73]. We added a few lines (295-300) in page 8 to explain the occurrence of point mutation with target gene expression.
10. The subsection 4 on Fusarium biofilms and azole resistance (pg 9-10, lines 323-409) primarily describes the life cycle of biofilms, in vivo models of Fusarium biofilm forming capacity and in vitro studies of biofilm composition. The above information should be shortened to one paragraph of 15 lines at the most, in order to give pertinent information on in vitro/in vivo studies regarding the antifungal activity of azoles against Fusarium
Response: Thanks for pointing this out. We agree that the subtitle “Fusarium biofilms and azole resistance” for this section was misleading. Our original intention was to summarize the biological and physiological characteristics of Fusarium biofilms in various in vitro and in vivo models, with the hope of identifying specific biofilm traits relevant to antibiofilm treatment strategies. From a clinical perspective, literature on Fusarium biofilms is quite limited, particularly in in vivo systems, making it challenging to draw direct parallels with plant-pathogenic systems. Due to the limited clinical data available, we had to rely heavily on studies of plant-associated Fusarium species in this review. Currently, the understanding of how fundamental biological features—including biofilms and genomic variations—affect drug susceptibility remains largely insufficient. To better reflect the scope of this section, we have revised the subtitle to "4. Fusarium Biofilms."